# Isotopic and microbotanical insights into Iron Age agricultural reliance in the Central African rainforest

Madeleine Bleasdale [1,2✉], Hans-Peter Wotzka[3], Barbara Eichhorn[4,10], Julio Mercader [1,5], Amy Styring[4,6], Jana Zech[1], María Soto[5], Jamie Inwood[5], Siobhán Clarke [5], Sara Marzo[1], Bianca Fiedler[1], Veerle Linseele[7], Nicole Boivin[1,5,8,9] & Patrick Roberts [1,8✉]

The emergence of agriculture in Central Africa has previously been associated with the migration of Bantu-speaking populations during an anthropogenic or climate-driven 'opening' of the rainforest. However, such models are based on assumptions of environmental requirements of key crops (e.g. *Pennisetum glaucum*) and direct insights into human dietary reliance remain absent. Here, we utilise stable isotope analysis ($\delta^{13}$C, $\delta^{15}$N, $\delta^{18}$O) of human and animal remains and charred food remains, as well as plant microparticles from dental calculus, to assess the importance of incoming crops in the Congo Basin. Our data, spanning the early Iron Age to recent history, reveals variation in the adoption of cereals, with a persistent focus on forest and freshwater resources in some areas. These data provide new dietary evidence and document the longevity of mosaic subsistence strategies in the region.

[1] Department of Archaeology, Max Planck Institute for the Science of Human History, Kahlaische Straße 10, 07745 Jena, Germany. [2] Department of Archaeology, University of York, King's Manor, Exhibition Square, York YO1 7EP, UK. [3] Institute of Prehistory, University of Cologne, Weyertal 125, 50931 Cologne, Germany. [4] Institute of Archaeological Sciences, Goethe University, Norbert-Wollheim-Platz 1, D-60629 Frankfurt am Main, Germany. [5] Department of Archaeology and Anthropology, University of Calgary, 2500 University Drive, N.W. Calgary, Alberta T2N 1N4, Canada. [6] School of Archaeology, University of Oxford, 1 South Parks Road, Oxford OX1 3TG, UK. [7] Department of Earth and Environmental Sciences, Center for Archaeological Sciences, University of Leuven, Celestijnenlaan 200E, 3001 Leuven, Belgium. [8] Department of Archaeology, University of Queensland, St Lucia QLD, 4072 Brisbane, Australia. [9] Department of Anthropology, National Museum of Natural History, Smithsonian Institution, 10th Street & Constitution Avenue, Washington, DC 20560, USA. [10]Deceased: Barbara Eichhorn. ✉email: bleasdale@shh.mpg.de; roberts@shh.mpg.de

For the past half a century, if not longer, the processes for the dispersal of Bantu-speaking communities from Western Central Africa have been a major focus of African archaeological, linguistic, and genetic research[1–4]. While there has been an increasing departure from notions of a single sweeping 'Bantu Expansion', the degree to which the movement of people, languages, and the emergence of farming are linked across Africa continues to be forcefully debated[5–7]. Central Africa is at a key location for developing existing models for the spread of farming[8] yet investigations of the emergence of food production, particularly in the rainforest, have been limited[9]. Assumptions that tropical rainforests represent substantial barriers to agriculturalists[10] have been used to rationalise a relatively late arrival of farming in the region, c. 2500 years ago, during a period of climate- or human-induced deforestation[11–13] (Supplementary Note 1). However, unlike other parts of Africa[14–16], there have been few studies directly testing changes in human dietary reliance on agricultural crops, relative to local freshwater, bushmeat, and tropical forest plant resources[17], from the first arrival of domesticates in the region through to the present day.

Linguistic, material culture, and radiocarbon analyses have now shown that human arrival throughout the Congo Basin was a complex and time-transgressive occurrence, potentially with the interaction of different populations occurring[10,18–25]. Furthermore, ideas relating to the inability of farming populations to occupy the tropical rainforests of Central Africa have come under renewed scrutiny[26]. Experimental research has demonstrated that pearl millet (Pennisetum glaucum) can be grown in forested portions of the Inner Congo Basin[27]. This suggests that discoveries of pearl millet (c. 2330–330 BP) at Iron Age sites across Central Africa, regions presently covered in tropical rainforest[18,28], need not represent a time of mass 'rainforest crisis'[29,30]. Not only that, but Iron Age expansions into the various tributaries of the Congo River continued well after the supposed peak in rainforest decline 2500 years ago, suggesting more complex, ongoing processes of agricultural adaptation, and settlement. Together, these developments make it essential to build more integrated, multidisciplinary, and context-specific insights into changes in diet and land use in different parts of Central Africa through time, as different agricultural populations negotiated their tropical surroundings.

Here, we present new, direct dietary information from Iron Age sites in the Democratic Republic of Congo (DRC) using the stable carbon ($\delta^{13}C$), nitrogen ($\delta^{15}N$), and oxygen ($\delta^{18}O$) isotope analysis of human and animal remains. Isotopic results were obtained for human burials from the sites of Imbonga (IMB; $n = 1$), Longa (LON; $n = 1$), Bolondo (BLD; $n = 18$), and Matangai Turu Northwest (MTNW; $n = 1$). In addition, bone collagen ($n = 10$) and enamel ($n = 6$) were analysed for a range of fauna from BLD to create an isotopic baseline. The sites studied represent different geographic and temporal contexts (Fig. 1). IMB is the type site for the earliest pottery tradition of the central equatorial rainforest and the individual analysed, indirectly dated to ~2050 BP (Supplementary Note 2), would have been a member of a group representing already established agriculture in the region after its initial settlement by sedentary immigrant populations a few centuries earlier. In contrast, individuals from LON and BLD represent subsistence practices during the Late Iron Age, when populations were spreading further across the Congo Basin. Finally, isotopic results from the individual from MTNW, previously identified as a likely hunter-gatherer[31], offers new evidence about the intricacies of subsistence, cultural, and genetic identities further to the eastern edge of the Basin (Supplementary Note 2). Collectively, the samples analysed span the period following the first arrival of food producers in this region (~2050 BP) through to relatively recent occupation (~130 BP;

Supplementary Note 2, Supplementary Tables 1 and 2, and Supplementary Figs. 1, 3–7).

$\delta^{13}C$ analysis of human tissues has long been demonstrated to provide insights into reliance on plants with different photosynthetic pathways (namely $C_4$ versus $C_3$) and their animal consumers (Supplementary Note 3)[32–34]. Significantly, in Central Africa, wild, as well as potentially domesticated (e.g., yams), forest plants are $C_3$, while incoming cereal crops (e.g., pearl millet, sorghum, and, for later periods, maize) are $C_4$. $\delta^{13}C$ measurements of wild plants and animals from the rainforests of the DRC show that these forests are largely composed of $C_3$ vegetation[35]. Moreover, they show a recognisable 'canopy effect' on this $C_3$ vegetation that results in lower $\delta^{13}C$ among plants, and their animal consumers, living under dense canopies compared to those living in more open areas[35], something that has been well-documented in many other tropical regions[34,36].

The sites of IMB, LON, and BLD are located on tributaries of the Congo River in the western DRC (Fig. 1 and Supplementary Note 2) an area presently covered in dense $C_3$-dominated evergreen and semi-deciduous forest[37]. Stable carbon measurements of faunal tooth enamel from BLD, which largely reflect the proportions of $C_3/C_4$ plants consumed, reflect local palaeoecology, as well as providing baseline values for human diet. The final site, MTNW, is situated in the closed-canopy forest of the Ituri Region of the Northeast Congo Basin with palaeoenviromental proxies, suggesting a predominance of tropical forest tree taxa during the time of occupation[38,39].

$\delta^{15}N$ analysis provides insights into the positions of humans within their trophic web and their potential consumption of aquatic resources[40,41], while $\delta^{18}O$ measurements reflect water sources and environments[42,43] (see "Methods" section or Supplementary Note 3 for full details). To the best of our knowledge, this is the first time that multi-tissue stable isotope analysis of prehistoric humans and animals has been applied in the Congo Basin, in order to provide long-term insights into changing human reliance on different resources. We also present results of $\delta^{13}C$ and $\delta^{15}N$ analysis of charred lumps interpreted as food residues from BLD, as well as microparticle analysis of dental calculus recovered from MTNW, to obtain more detailed insights into prepared and consumed foods, respectively.

## Results

**Faunal and human bone collagen.** Bone collagen results were assessed using established indicators of preservation, including a C/N ratio between 2.9–3.6, %C of ca.15–48%, and %N of ca. 5–17% (ref. [44–47]). Two results generated for human burials from BLD were excluded from final analysis as they produced a %N < 5 and a %C < 15 % (Supplementary Table 3).

The fauna from BLD fall broadly into four groups: wild browsers (antelope and duiker, $n = 2$), domesticated browsers (goats, $n = 2$), mammalian carnivores ($n = 2$), and aquatic species (fish and crocodile, $n = 4$). $\delta^{13}C$ values from BLD mammals ($n = 6$) are consistent with a largely $C_3$-based diet with measurements ranging from −23.7 to −18.2‰, although the goats, the dog, and the fox-sized carnivore could potentially have some $C_4$ component to the diet (Fig. 2 and Supplementary Table 3). The common duiker (Sylvicapra grimmia, BLD 83/2-8) and small antelope (BLD 83/2-6 + 7) have $\delta^{15}N$ values (5.9 and 7.3‰, respectively) consistent with herbivorous diets. In contrast, the fox-sized carnivore (13.7‰) and dog (12.3‰) display higher $\delta^{15}N$ that is consistent with consumption of animal protein. Three of the aquatic species sampled had higher $\delta^{15}N$ values than the catfish ($\delta^{15}N$ 9.4‰), the crocodiles gave $\delta^{15}N$ of 11.1 and 11.4‰, and the bichir (Polypterus sp.), a fish known to consume other fish and small vertebrates[48], produced a $\delta^{15}N$ measurement of 12.7‰.

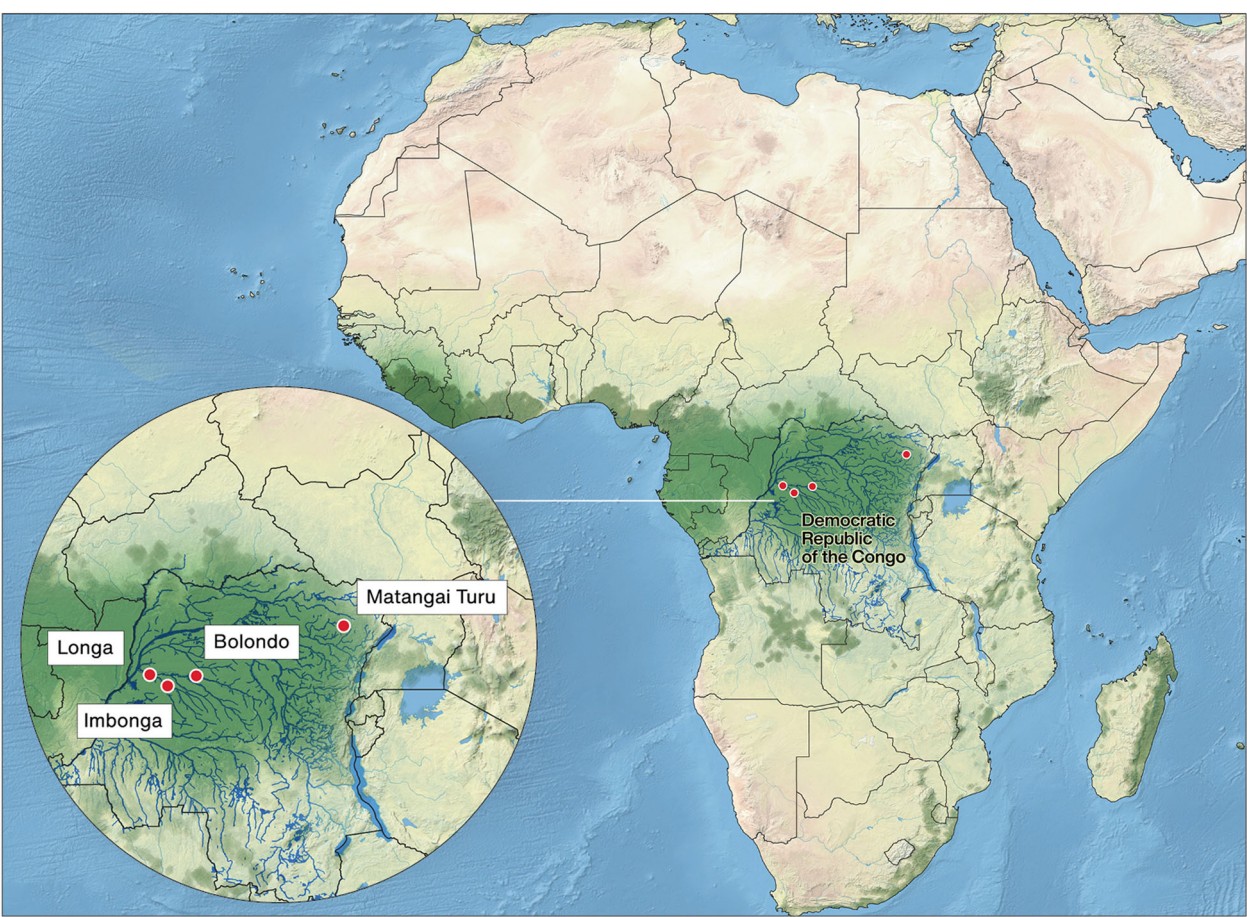

**Fig. 1 Map showing the location of the archaeological sites of study in the Democratic Republic of the Congo.** Imbonga (IMB), Longa (LON), Bolondo (BLD), and Matangai Turu Northwest (MTNW). Tropical rainforest is shown in dark green. The map was created for this study by Hans Sell (Graphic Designer for the Max Planck Institute for the Science of Human History, Jena, Germany) using QGIS 3.12 https://qgis.org/en/site/ and the Natural Earth Database from https://www.naturalearthdta.com/downloads/. To increase accuracy, river locations are based on OpenStreetMap data provided by GEOFABRIK http://download.geofabrik.de/africa/congo-democratic-republic-latest-free.shp.zip. Final adjustments to colour saturation and site labels were made using Adobe Illustrator and Photoshop.

The accompanying $\delta^{13}C$ value of $-24.7‰$ for the bichir somewhat overlaps with that expected for $C_3$ plants and animals, demonstrating the need to examine both $\delta^{15}N$ and $\delta^{13}C$, in order to separate freshwater fish and forest resources.

The $\delta^{13}C$ values for the bone collagen of humans from BLD ($n = 11$) dating to between 1426 and 1942 years cal. AD (Supplementary Figs. 3–7) are quite variable, ranging from $-21.0$ to $-16.3‰$. Alongside the individual from LON (directly dated to 1642—after 1938 cal. AD), these values are generally consistent with reliance on $C_3$ plants, $C_3$ plant-consuming wild and domestic animals, and freshwater resources. The average $\delta^{15}N$ value of 14.5‰ for the human individuals from BLD in comparison to the average for the goats (9.9‰) is within the range of values reported for diet–collagen spacing, potentially indicating reliance on these domesticates[41,49]. However, $\delta^{15}N$ values ranging from 13.4 to 16.9‰, and three individuals with $\delta^{15}N$ values higher than 15‰, as well as the riverine setting and modern and historical evidence that the site was a fishing camp (Supplementary Note 2), indicate that freshwater fish was also a major part of human diets at BLD. Nevertheless, it is still evident that all humans and domestic animals, as well as the fox-sized carnivore, have higher $\delta^{13}C$ values than the available wild $C_3$ or freshwater fauna, indicating the consumption of an additional resource enriched in $\delta^{13}C$.

A visible negative correlation between human $\delta^{15}N$ and $\delta^{13}C$ suggests that this was, in fact, a plant food (Fig. 2), although a Pearson's rank test (correlation coefficient = $-0.600$, d.f. = 9, $p = 0.05$) produced results at the limit of statistical significance. This is likely a product of the small sample sizes available. The metabolic routing bias of bone collagen $\delta^{13}C$ towards protein components of the diet (at the expense of carbohydrate and fat inputs), and importance of high protein freshwater fish in diets of the measured individuals, means that we can expect that consumption of this low-protein plant resource was actually greater than it appears in the bone collagen values of Fig. 2. Thus, although somewhat underrepresented in the present bone collagen isotope results, there is a signal indicative of some $C_4$ plant component, which must have contributed to the diet of Late Iron Age humans and domestic animals in the Inner Congo Basin from at least the 15th century cal. AD onwards.

**Faunal and human tooth enamel.** Tooth enamel is widely regarded as the archaeological material of choice in the tropics. Stable carbon and oxygen isotopes of tooth enamel have been shown to robustly preserve ecological variation, even in tropical regions, from the Miocene to the Late Pleistocene[34,50,51]. Tooth enamel was sampled from human second and third molars

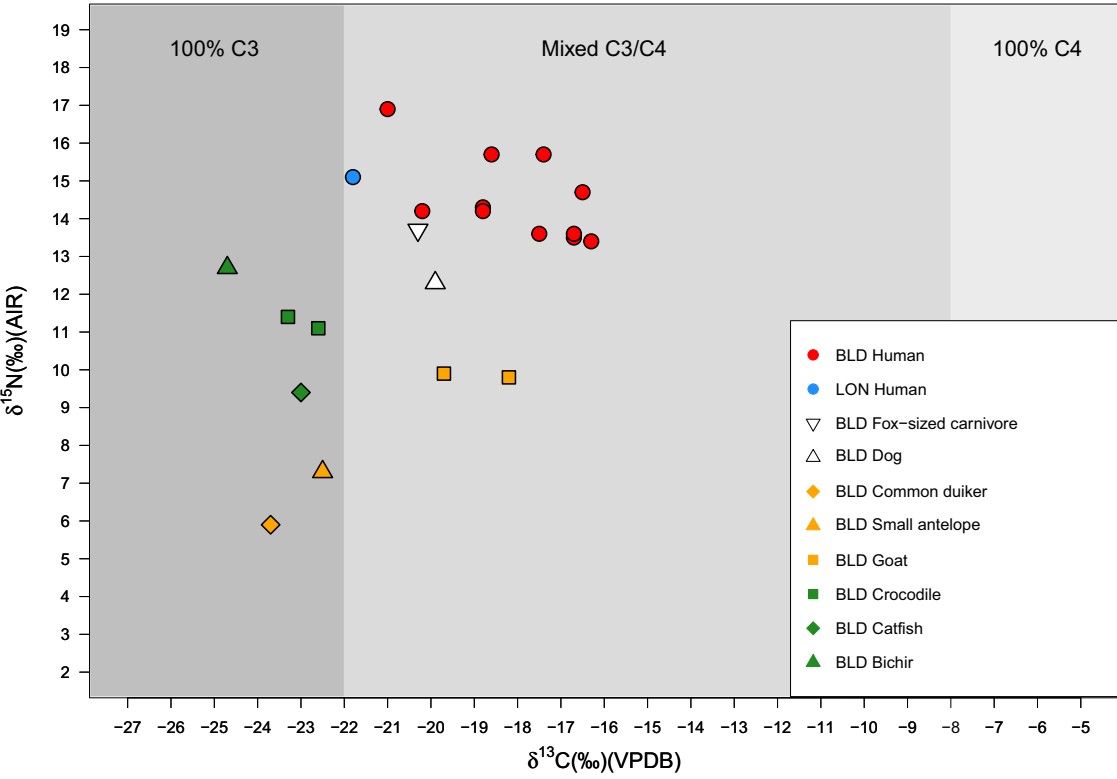

**Fig. 2 Human and faunal bulk bone collagen δ¹³C and δ¹⁵N results for BLD and LON.** Shading indicates estimated bone collagen δ¹³C for individuals consuming 100% C₃, mixed C₃/C₄, and 100% C₄ sources[33].

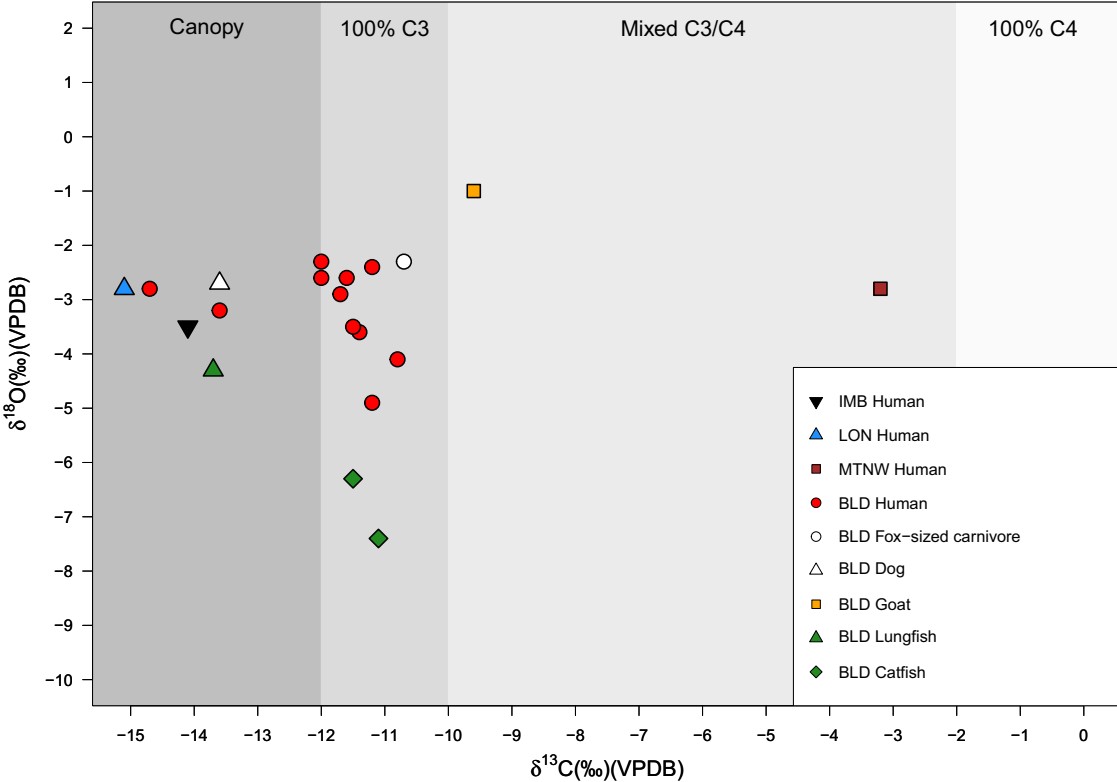

**Fig. 3 Human and faunal bulk tooth enamel δ¹³C and δ¹⁸O for BLD, IMB, LON, and MTNW.** Shading indicates estimated tooth enamel δ¹³C for individuals living under dense canopy and consuming 100% C₃, mixed C₃/C₄, and 100% C₄ sources based on literature[34,51].

enabling the investigation of diet during late childhood–early adolescence. Furthermore, given that tooth enamel $\delta^{13}C$ reflects the isotopic composition of the whole diet (including carbohydrates, fats, and proteins), as opposed to the dominance of the protein signal in bone collagen, tooth enamel $\delta^{13}C$ enables additional dietary resolution (see Supplementary Note 3).

Faunal and human $\delta^{13}C$ enamel results from IMB, LON, BLD, and MTNW (Fig. 3 and Supplementary Table 4) reveal diverse subsistence strategies that can be divided into three broad groups: wild forest resources, forest and freshwater resources, and more open environments, including $C_4$ food sources. The individual from the earliest site mentioned here, IMB, has a $\delta^{13}C$ value of −14.1‰ indicating reliance on a mixture of tropical rainforest and freshwater resources, while the individual from LON shows clear evidence for reliance on tropical rainforest resources with a $\delta^{13}C$ value of −15.1‰. Two individuals from BLD gave $\delta^{13}C$ values of −14.7 and −13.6‰ also suggesting a heavy reliance on tropical or freshwater resources. The $\delta^{13}C$ values of remaining individuals from BLD (−12.0 to −10.8‰) are indicative of a dominance of $C_3$ food sources, possibly yams, plantain, or oil palm grown in slightly more open conditions[52,53], or perhaps a mixed diet of closed canopy and freshwater resources. There is no clear evidence for dietary reliance on $C_4$ plants in any of the human tooth enamel samples analysed from IMB, LON, or BLD.

As enamel carbonate is reflective of all food sources it would not be affected by a high degree of fish consumption, making this lack of evidence for $C_4$ consumption somewhat surprising given the collagen data. For four individuals from BLD, it was possible to analyse both bone collagen and tooth enamel to investigate tissue-specific or age-related dietary differences. One individual (BLD 83/3 individual 2) demonstrates a predominant reliance on $C_3$ or freshwater food sources from mid-late childhood through to adulthood giving a $\delta^{13}C_{enamel}$ of −11.7 and $\delta^{13}C_{coll}$ value of −21.0‰. Interestingly, however, BLD 83/1 individual 1 $\delta^{13}C_{enamel}$ value (−13.6‰) is consistent with a reliance on tropical rainforest and freshwater resources, but the accompanying $\delta^{13}C_{coll}$ value (−17.4‰) reflects a $C_3/C_4$-based protein diet. A similar shift is also seen when comparing enamel and collagen results for BLD 83/8 ($\delta^{13}C_{enamel}$ −11.4‰, $\delta^{13}C_{coll}$ −20.2‰) and BLD 83/10 ($\delta^{13}C_{enamel}$ −10.8‰, $\delta^{13}C_{coll}$ −16.7‰). These distinctions could be a product of either (i) the fact that low-protein rainforest plant resources are more visible in the tooth enamel, while higher-protein animals eating $C_4$ plants (e.g., goats) or higher $\delta^{13}C$ freshwater resources, with higher $\delta^{13}C$ values are more visible in collagen or (ii) greater consumption of $C_4$-based resources in adulthood compared to childhood.

In sharp contrast to the western DRC samples, the M3 from the individual from MTNW has a $\delta^{13}C$ enamel value of −3.2‰ that clearly indicates that $C_4$ resources made the dominant contribution to the diet of this individual. Unfortunately, it was not possible to analyse bone collagen from this individual, but multiple teeth (M1–M3) were analysed to investigate diet throughout childhood. The results (Supplementary Fig. 8) demonstrate that this individual relied upon $C_4$ food sources throughout childhood and as a juvenile.

**Microbotanical remains dental calculus.** Dental calculus was analysed from three mandibular molars (M1, M2, and M3) from the MTNW individual and a total of 38 starch granules and 9 phytoliths were retrieved (Supplementary Table 5). Micro-charcoal is very common (Fig. 4a, aq–as). The calcium phosphate matrix was decontaminated prior to decalcification, as per a published protocol[54] in which calculus is immersed in sodium hydroxide of 2% w/v solution for 24 h. As expected for ancient starch granules, the discovered calculus starch displays signs of

damage to their semicrystalline matrix, having partially or totally lost their native birefringence. Other signs of diagenesis include fissuring, pitting, granulation, and implosion of the hilum. The taxonomic identification of starch granules depends on whether they represent unique morphometric identifiers that can be compared to published reference collections. In this respect, the mixture of polygonal, orbicular, and quadratic granules found derive from a grass seed ($n = 26$), but cannot specifically be assigned to pearl millet (*Pennisetum glaucum*), wild finger millet, (*Eleusine africana*), or domestic finger millet (*Eleusine coracana*), as they all have in common compound granules and/or markedly polygonal shapes with mean metrics <10 μm (Supplementary Fig. 9a–c).

In contrast, starch granules from *Sorghum bicolor* are polymorphic with roughly polygonal, orbicular, and quadratic shapes, a centric hilum often creased or slit (Supplementary Fig. 9d–h), and can be uniquely identified by prismatic–polygonal shapes with mean maximum length 18–30 μm (ref. [55]). Another common starch granule type identified is parabolic and/or oblong elongate ($n = 11$; Fig. 4). In our reference collection, the best possible match for this cohort is in the Dioscoreaceae, which in sub-Saharan Africa produces the highest number of unique identifiers in granule morphometrics and overwhelmingly associates with wild yams[55].The remaining type (ovate, $n = 1$; Fig. 4) is also tentatively associated with an underground storage organ, and similar granules occur in the Asphodelaceae family[55]. With regards to phytoliths, all phytoliths ($n = 9$) come from one tooth (M1) and they are characterised as medium to large, brown globular bodies, with tuberculate to echinate projections. These large, brown phytoliths with variably tuberculated to echinated spines are referenced in the nutshell of *Elaeis guineensis* (Supplementary Fig. 9i–n), whose charred remains were also discovered at the site.

**Charred food fragments from Bolondo.** The charred food fragments recovered from contexts at BLD ($n = 8$) fall into three groups based on their $\delta^{13}C$ values: those with $\delta^{13}C$ values < −27‰; a single food fragment with an intermediate $\delta^{13}C$ value of −24‰; and two food fragments with $\delta^{13}C$ values around −9‰ (Supplementary Table 6). $\delta^{13}C$ values ranging from < −27 to −24‰ are likely indicative of food fragments consisting of $C_3$ or aquatic resources, while $\delta^{13}C$ values of −9‰ indicate that the primary content was likely $C_4$ resources. The charred food fragments from BLD contained between 0.6 and 2.5% N, which means that for all samples apart from BE06, the $N_2$ peak was too small for reliable determination of their $\delta^{15}N$ values. BE06, a sample with a high $\delta^{13}C$ value (−9.3‰), has a relatively high $\delta^{15}N$ value (7.8‰) compared to those of herbivores from the site, perhaps indicating that $C_4$ plant resources like pearl millet and sorghum consumed by humans were growing in different soil conditions compared to the plants eaten by wild and domestic herbivores.

## Discussion

The discovery of pearl millet *(Pennisetum glaucum)* in Iron Age pits in Southern Cameroon[18] was one of the most significant findings of the past two decades in Central African archaeology. The discovery sparked debate over the environmental context for early agriculture[26,56], contributing to an increasingly complex narrative for the settlement of the Central Africa rainforest[10,21,57,58]. Yet direct assessments about the degree to which these early farming communities, particularly in the DRC, relied on $C_4$ resources are limited[17]. Our data enable direct exploration of the adoption of agriculture at different points during the Iron Age in the DRC and highlight substantial regional

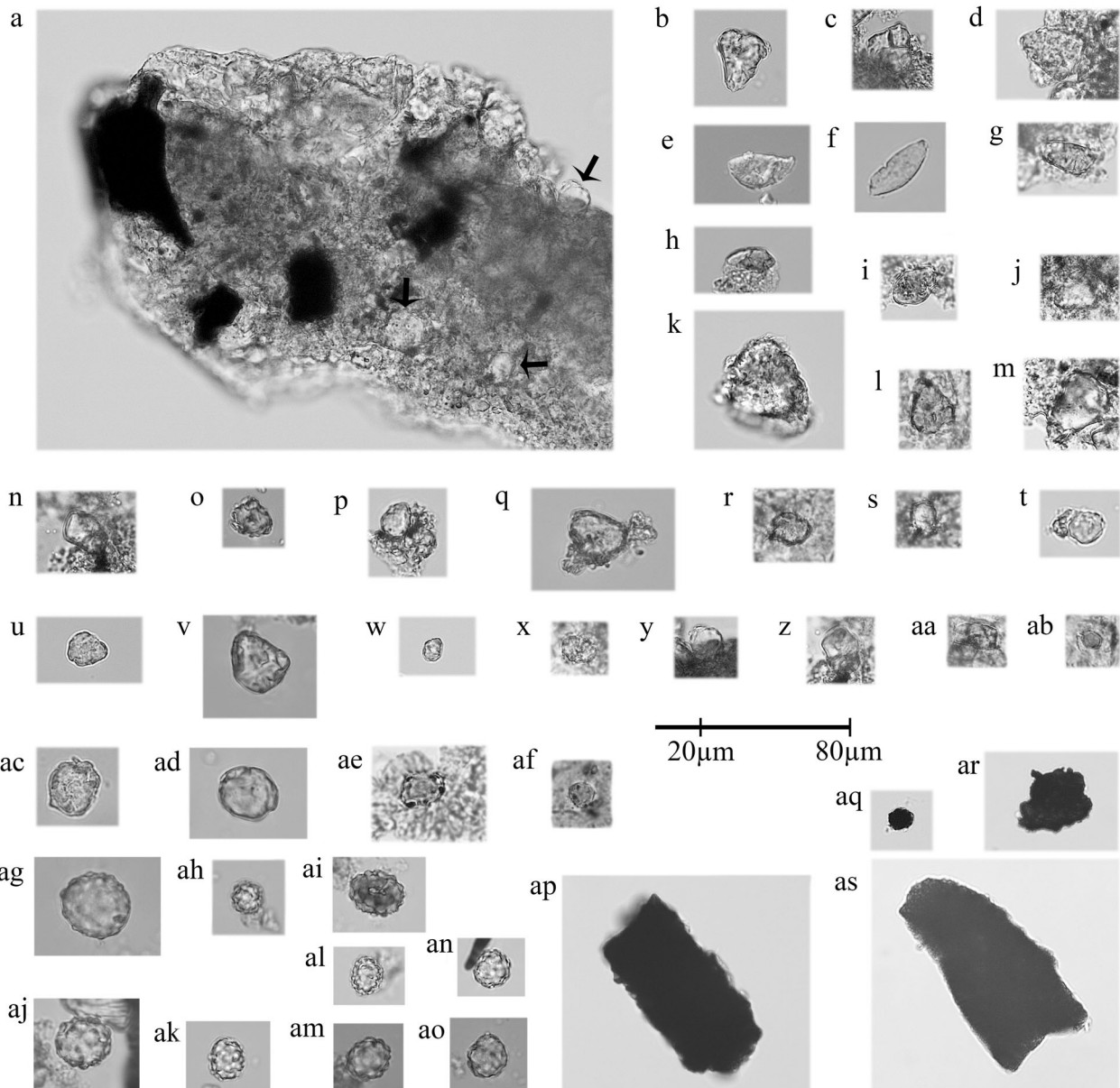

**Fig. 4 Selected microbotanical materials extracted from the dental calculus of MTNW. a** Starch and microcharcoal entrapped within calcified matrix, M2. Starch granules: **b–d** parabolic, M2; **e** oblong elongate, M2; **f** oblong elongate, M3; **g, h** parabolic, M2; **i** ovate, M2; **j–m** parabolic, M2; **n** quadratic, M2; **o** polygonal, M2; **p** orbicular, M3; **q** polygonal, M3; **r** orbicular, M2; **s** quadratic, M2; **t** polygonal, M2; **u** orbicular, M2; **v** polygonal, M1; **w** orbicular, M3; **x** orbicular, M2; **y** orbicular, M2; **z** polygonal, M2; **aa** quadratic, M2; **ab** orbicular, M2; **ac** polygonal, M2; **ad** orbicular, M1; **ae** orbicular, M2; **af** orbicular, M2; phytoliths: ag–ao globular tuberculate to echinate, M1; microcharcoal: **ap–as** M2.

variability, particularly with regards to uptake of $C_4$ crops into the diet. The IMB individual, indirectly dated to ~2050 BP (Supplementary Note 2), provides a snapshot of potentially early agricultural groups entering the region. While this period is commonly associated with the arrival of cereal cultivation, this individual shows a heavy reliance on $C_3$ closed rainforest or freshwater resources rather than $C_4$ crops. Moving to the Late Iron Age and ongoing 'Bantu expansion' into a number of tributaries of the Congo River, individuals from LON and BLD show no clear evidence for a dietary reliance on $C_4$ plants in tooth enamel samples. Collagen values, however, do suggest some degree of $C_4$ consumption in addition to a core reliance on $C_3$ closed rainforest and more open $C_3$ resources, perhaps including plantain, oil palm and yams or manioc, and freshwater resources. This interpretation is supported by zooarchaeological research at

BLD that highlights a dominance of fish, as well as the apparent importance of riverine locations for these settlements.

These findings are particularly interesting given that there is clear archaeobotanical evidence, at BLD in particular, for the presence of pearl millet. $\delta^{13}C$ data from two charred food fragments from flotation samples from BLD, as well as identified charred pearl millet grains, definitively show that millet was processed in this western portion of the DRC during the Iron Age. However, it did not apparently dominate the diets of buried individuals at the same site, though its exact importance in adulthood is somewhat masked by protein representation of bone collagen. This potentially implies that millet was used in a different context in the Iron Age of this region and, instead of being a staple, was possibly used in feasting, brewing, or prestige contexts[59]. Such an interpretation could be supported by the fact that

multi-tissue $\delta^{13}C$ analysis of humans at BLD indicates increased $C_4$ contributions in adult life relative to childhood or juvenile diets, though this remains tentative at present. Regardless, results encourage a shift away from broad linguistic and genetic models for the 'Bantu expansion' when studying agricultural adaptations in Central Africa, and necessitates further direct, context-specific multidisciplinary analyses in order to understand the adoption and incorporation of $C_4$ cereal crops in Iron Age subsistence in the western DRC. An experimental study has demonstrated that it is possible to grow pearl millet within the Inner Congo Basin today[27]. While pollen evidence suggests the existence of a 'rainforest crisis' c. 2500 years ago[60,61], this study and our data demonstrate that this was not necessary for the initial, or indeed subsequent, cultivation of millet in the region.

The importance of undertaking a multidisciplinary, contextual approach to the emergence of food production in Central Africa is further indicated by the data from the northeastern DRC site of MTNW. In contrast to the western DRC sites, the stable isotopic data from this individual highlight a clear overall dietary reliance on $C_4$ food sources throughout childhood and into teenage years. The location of Matangai Turu in proximity to migrating farming populations in eastern Africa[62], observed genetic affinity of the sampled individual to these groups[63], as well as hunter-gatherer populations[63], and the identification of Poaceae starch (likely sorghum) in the dental calculus of this individual indicate that this $C_4$ signal represents the use of sorghum possibly acquired through interactions with agricultural groups in eastern Africa. Nevertheless, the human remains date to $813 \pm 35$ BP, a time when the region is believed to have been covered by lowland Guineo-Congolain rainforest[31,35,38,64]. Moreover, the presence of starches from Dioscoreaceae and Asphodelaceae in the dental calculus, as well as the association with wild rainforest fauna[54,64] indicate ongoing contributions of forest resources to the lifeways of this Late Iron Age population. Evidently, the adoption of cereal crops into subsistence economies across tropical Africa was not uniform, displaying regional variation and incorporation into existing, dynamic lifeways.

The results of this study reveal a diversity of subsistence practices spanning the Central African Iron Age that involved the incorporation of incoming $C_4$ crops to varying degrees. For the Matangai Turu individual this likely reflects complex forager–farming interactions and exchanges during this time, which have been increasingly recognised in genetic studies[65,66]. In contrast, there is presently no evidence for pre-Iron Age indigenous settlement in the Inner Congo Basin, though the local agency of the Late Iron Age populations adapting to the western portion of the region should not be underestimated. Tropical forests in the DRC, as elsewhere[67], were home to diverse groups who developed a range of strategies to procure and produce food. We can only begin to look at these dynamic, contextually dependent strategies if we move away from sweeping narratives based on genetic or linguistic data to focus on direct evidence from archaeology, archaeobotany, archaeozoology, and biomolecular methodologies relating to the actual significance of different resources to diets across time and space.

In this way, we can also begin to properly understand the wider significance of ongoing agricultural adaptations in Central Africa. The Congolese portions of the Central African rainforest are considered some of the most vulnerable to climate change[68–70] and are increasingly appreciated as essential, but now threatened, carbon 'sinks' for the continental and global carbon cycle[37,71]. Consequently, there has been substantial discussion about the relative sustainability and antiquity of intensive agricultural land use[72,73] versus mixed agricultural, agroforestry, and hunting strategies in the region[74,75]. Despite frequent NGO or government calls to focus on productive cereal monoculture to meet

growing populations in West and Central Africa[76], it is clear that mixed use of $C_4$ plants, wild resources, rainforest economic plants, and freshwater resources have characterised subsistence practices in the western DRC for over 2000 years. Indeed, the continuation of heterogeneous food production strategies could prove crucial in ensuring long-term food security in tropical Central Africa, as well as the survival of Congolese environments crucial to the global carbon cycle[77] pan-African precipitation[78], and global biodiversity[79].

## Methods
Four Iron Age sites were selected for study from across the DRC (Fig. 1 and Supplementary Note 2): IMB, LON, BLD, and MTNW. Bulk bone collagen $\delta^{13}C$ and $\delta^{15}N$ measurements were obtained for human burials from LON ($n=1$), BLD ($n=11$), and MTNW ($n=1$). Totals reflect results taken forward for interpretation. For BLD, two results were excluded from final analysis due to poor preservation (for full details see: "Results" section and Supplementary Table 3). Bone collagen was also analysed from a range of faunal remains from BLD ($n=10$) to establish a dietary baseline, including domestic browsers (goats), wild browsers (duiker), and fish. To further explore dietary intake, human tooth enamel was sampled from IMB ($n=1$), LON ($n=1$), BLD ($n=11$), and MTNW ($n=1$). Due to differential preservation, it was only possible to generate both a bulk collagen and tooth enamel results for four human burials from BLD, and the single individual from LON. In addition, animal tooth enamel from BLD ($n=6$) was analysed for $\delta^{13}C$ and $\delta^{18}O$ to aid the interpretation of human values.

BLD is a c. 660 BP to present site located in the western Interior Congo Basin on the floodplain of the Tshuapa River. The site was first excavated in 1983 with the most recent field season taking place in 2016, with financial support of the Deutsche Forschungsgemeinschaft[80]. Owing to partly waterlogged conditions, the level of organic preservation at BLD is far higher than at sites located above the floodplain, and a number of human burials have been excavated[80] (Supplementary Fig. 2). In addition, tooth enamel samples were analysed from one human individual from each of the sites of IMB and LON. Both sites were excavated by Manfred Eggert in the 1980s. IMB is the type site for the oldest pottery of the equatorial forest and is located on the Momboyo River, and LON is located on the Ruki River[20,81].

Finally, a single individual dating to $813 \pm 35$ BP was analysed from MTNW, a Later Stone Age rockshelter located in the Ituri rainforest of the Eastern Congo Basin[31]. This individual was previously identified as a likely hunter-gatherer based on morphological evidence, associated lithics, presence of wild fauna, and absence of domesticated plants[31]. However, the presence of a large assemblage of ceramics, iron slag, and iron objects means it is impossible to say definitively whether the individual was from a primarily foraging group or associated with Bantu-speaking groups[31]. Unfortunately, it was not possible to sample any associated fauna from MTNW for this study, but a range of forest taxa were present at the site, including porcupines, antelopes, primates, small bovids, and snails[64]. While there are clear geographical and ecological differences between MTNW and BLD the fauna at both sites are indicative of a closed forest environment. For the individual from Matangai Turu, it was possible to sample all three permanent molars to track dietary consumption throughout childhood. The tooth enamel $\delta^{13}C$ of human molars is influenced by dietary intake during the time of tooth formation with the first molar forming between 2 months prior to birth to 4 years after birth, the second molar between 4 and 7 years, and third molar between 9 and 16 years (Supplementary Note 3).

**Stable isotope analysis of bone collagen.** Human and faunal bone collagen was extracted using a modified Longin[82] method. Bone samples (~1 g) were broken into small pieces and adhering soil was removed by abrasion using a sandblaster. Samples were demineralised by immersion in 0.5 M HCl for 1–7 days. Once demineralisation was complete, samples were rinsed three times with ultra-pure $H_2O$. The residue was gelatinised in pH 3 HCl at 70 °C for 48 h, and the soluble collagen solution Ezee-filtered to remove insoluble residues[83]. Samples were lyophilised in a freeze dryer for 48 h. Where sufficient material was available, ~1.0 mg of the resulting purified collagen was weighed, in duplicate, into tin capsules for analysis.

The $\delta^{13}C$ and $\delta^{15}N$ ratios of the bone collagen were determined using a Thermo Scientific Flash 2000 Elemental Analyser coupled to a Thermo Delta V Advantage mass spectrometer at the Isotope Laboratory, MPI-SHH, Jena. Isotopic values are reported as the ratio of the heavier isotope to the lighter isotope ($^{13}C/^{12}C$ or $^{15}N/^{14}N$) as $\delta$ values in parts per mill (‰) relative to international standards, VPDB for $\delta^{13}C$ and atmospheric N2 (AIR) for $\delta^{15}N$. Results were calibrated against international standards of (IAEA-CH-6: $\delta^{13}C = -10.80 \pm 0.47$‰, IAEA-N-2: $\delta^{15}N = 20.3 \pm 0.2$‰, and USGS40: $\delta^{13}C = -26.38 \pm 0.042$‰, $\delta^{15}N = 4.5 \pm 0.1$‰) and a laboratory standard (fish gelatin: $\delta^{13}C = \sim -15.1$‰, $\delta^{15}N = \sim 14.3$‰). Based on replicate analyses long-term machine error over a year is $\pm 0.2$‰ for $\delta^{13}C$ and $\pm 0.2$‰ for $\delta^{15}N$. Overall measurement precision was studied through the measurement of repeats of fish gelatin ($n=80$, $\pm 0.2$‰ for $\delta^{13}C$ and $\pm 0.2$‰ for $\delta^{15}N$). The faunal ($n=8$) and human ($n=12$) bone collagen results from BLD and

LON are presented in Supplementary Table 3. Samples with a C/N ratio between 2.9–3.6, %C of ca.15–48, and %N of ca.5–17% were carried forward for interpretation[44–46].

**Stable isotope analysis of tooth enamel.** Approximately 10 mg of tooth enamel powder was obtained from sampled teeth using gentle abrasion with a diamond-tipped drill along the full length of the buccal surface and transferred to a microcentrifuge tube. Teeth were sampled from both humans and fauna (where available) across the four sites. Second and third molars were preferentially sampled from humans providing a long-term insight into juvenile diet, and avoiding the weaning effect potentially visible in first molars. Tooth enamel samples were pretreated with 1 mL 1% NaClO for ~60 min. The samples were rinsed three times with ultra-pure $H_2O$ and centrifuged before 1 mL 0.1 M acetic acid was added for 10 min. After this, samples were rinsed with ultra-pure $H_2O$, for a total of three washes[84,85]. After the final rinse, each tube was placed in a freeze drier for 4 h. In addition, an in-house standard of equid tooth enamel was processed alongside the samples of this study. Approximately 2 mg of the pretreated sample was weighed out into 12 mL borosilicate glass vials for analysis.

Following reaction with 100% phosphoric acid at 70 °C, sample $CO_2$ evolved and was analysed for stable carbon ($^{13}C/^{12}C$) and oxygen isotopic ratio ($^{18}O/^{16}O$) composition using a Thermo Gas Bench 2 connected to a Thermo Delta V Advantage Mass Spectrometer. Carbon ($\delta^{13}C$) and oxygen ($\delta^{18}O$) stable isotope values were calibrated against international standards IAEA NBS 18 ($\delta^{13}C$ −5.014 ± 0.032‰, $\delta^{18}O$ −23.2 ± 0.1‰), IAEA 603 ($\delta^{13}C$ +2.46 ± 0.01‰, $\delta^{18}O$ −2.37 ± 0.04‰), IAEA CO8 ($\delta^{13}C$ −5.764 ± 0.032‰, $\delta^{18}O$ −22.7 ± 0.2‰), and USGS44 ($\delta^{13}C$ = ~−42.1‰) registered by the International Atomic Energy Agency. Machine error based on the analyses of standards is ±0.1‰ for $\delta^{13}C$ and ±0.2‰ for $\delta^{18}O$. Overall measurement precision was assessed through repeat measurements of MERCK $CaCO_3$ ($n = 20$, ±0.2‰ for $\delta^{13}C$ and ±0.2‰ for $\delta^{18}O$, $\delta^{13}C$ = ~ −40.6‰, $\delta^{18}O$ = ~−13.3‰) and an in-house equid tooth standard ($n = 10$, ±0.3‰ for $\delta^{13}C$ and ±0.2‰ for $\delta^{18}O$).

**Microparticle analysis of dental calculus from MTNW.** Dental calculus was processed from three mandibular molars (M1, M2, and M3) from the MTNW individual. Images of the mineralised plaque prior to removal from the teeth, as well as those from contaminant starch granules and phytoliths are published elsewhere (see ref. [54]; Fig. 2). The elemental breakdown includes carbon, oxygen, calcium, and phosphorus, with small quantities of aluminium, silicon, nitrogen, sodium, and chlorine[54], and the Ca:P ratio was 2:1–1:7 indicating hydroxyapatite. We present microbotanical materials released from the calcified matrix after thorough decontamination protocols and decalcification in a cleanroom laboratory[54], as well as microbotanicals still trapped in the calculus matrix, but visible enough to have their two dimensional morphology identified. Identifications were made according to published morphometric classification criteria for the identification of ancient starch from sub-Saharan plants[55].

**Stable isotope analysis of charred food fragments.** Charred fragments classified as prepared food remains during archaeobotanical analysis at BLD were retrieved from flotation samples after sorting under a binocular microscope. A total of 2–3 mg of each sample was weighed into tin capsules for stable carbon and nitrogen isotope analysis. The $\delta^{13}C$ and $\delta^{15}N$ ratios of the charred food fragments were determined, using a Thermo MAT 253 continuous flow isotope ratio mass spectrometer coupled to a Thermo Flash 1112 Series elemental analyser in the Institut für Geowissenschaften, Goethe-Universität, Frankfurt am Main, Germany. Isotopic data are provided in Supplementary Table 6.

The carbon contents of the samples were calculated based on the area under the $CO_2$ peak relative to the weight of the sample, calibrated using IAEA-CH-7. Stable carbon isotope values were calibrated to the VPDB scale using IAEA-C-7 ($\delta^{13}C$ −32.15 ± 0.05‰) and IAEA-USGS24 ($\delta^{13}C$ −16.05 ± 0.04‰). Measurement uncertainty in $\delta^{13}C$ values was monitored using three in-house standards: LEU (DL-leucine, $\delta^{13}C$ −28.3 ± 0.1‰), GLU (DL-glutamic acid monohydrate, $\delta^{13}C$ −10.4 ± 0.1‰), and MIL (millet flour from a single panicle from a plot in Senegal, $\delta^{13}C$ −10.2 ± 0.1‰; Supplementary Data 1). Precision ($u(R_w)$) was determined to be ±0.06‰, accuracy or systematic error ($u(bias)$) was ±0.11‰, and the total analytical uncertainty in $\delta^{13}C$ values was estimated to be ±0.13‰, using the equation presented in Supplementary material (Supplementary Data 1).

The nitrogen contents of the samples were calculated based on the area under the $N_2$ peak relative to the weight of the sample, calibrated using IAEA-N2. Stable nitrogen isotope values were calibrated to the AIR scale using IAEA-N-1 ($\delta^{15}N$ 0.4 ± 0.2‰) and IAEA-N-2 ($\delta^{15}N$ 20.3 ± 0.2‰). Measurement uncertainty in $\delta^{15}N$ values was monitored using three in-house standards: LEU (DL-leucine, $\delta^{15}N$ 6.5 ± 0.4‰), GLU (DL-glutamic acid monohydrate, $\delta^{15}N$ −1.9 ± 0.1‰), and MIL (millet flour from a single panicle from a plot in Senegal, $\delta^{15}N$ 3.1 ± 0.6‰). $u(R_w)$ was determined to be ±0.18‰, $u(bias)$ was ±0.59‰, and the total analytical uncertainty in $\delta^{15}N$ values was estimated to be ±0.61‰.

**AMS dating.** Bone samples from five individuals from BLD, and the single individual from LON were sent for radiocarbon dating at the Scottish Universities Environmental Research Centre AMS Laboratory, Glasgow (SUERC, Lab ID: GU), in order to improve understanding of their chronology. Samples were pretreated using previously published methods[86]. Radiocarbon ages were calibrated to calendar timescale using OxCal 4 (ref. [87]) and IntCal13 atmospheric calibration curve[88] (Supplementary Note 2 and Supplementary Figs. 1, 3–7).

**Statistics and reproducibility.** Sample size was determined by archaeological sample preservation and availability. Data for human bone collagen were analysed using Pearson's $r$ R (version 3.5.3).

**Reporting summary.** Further information on research design is available in the Nature Research Reporting Summary linked to this article.

## Data availability

All of the data reported in the paper are presented in the main text or in the Supplementary Information, Tables, and Figures. Human and faunal skeletal samples have the following site-specific prefixes: 'IMB' for Imbonga, 'BLD' for Bolondo, 'LON' for Longa, and 'MTNW' for Matangai Turu Northwest. All human and faunal sample IDs are provided in Supplementary Tables 3 and 4. Charred food remains from BLD have sample codes starting 'BE' and full IDs are provided in Supplementary Table 6. The majority of individuals analysed in this study are from BLD and were excavated in 1983 by Manfred Eggert (Supplementary Note 2, and Supplementary Tables 3 and 4). Individuals from LON and IMB were also excavated by Manfred Eggert in the 1980s (Supplementary Note 2). The individual from MTNW was excavated by Julio Mercader and colleagues in the late 1990s (Supplementary Note 2). Additional faunal remains from 2016 excavations at BLD were selected for study based on species identifications (Supplementary Tables 3 and 4) and degree of preservation. Human and faunal skeletal material remaining from the stable isotope analyses, from individuals excavated at BLD, LON, and IMB, are currently housed at the Stable Isotope laboratory, Department of Archaeology, Max Planck Institute for the Science of Human History, Jena, Germany. Charred food remains from BLD are currently stored at the Institute for Archaeological Sciences, Goethe University, Frankfurt am Main, Germany. The skeletal remains of the individual from MTNW are housed at the Department of Biological Sciences, Complutense University of Madrid, Madrid, Spain. Dental calculus from the same individual is stored at the Department of Anthropology and Archaeology, University of Calgary, Canada. All data supporting the findings of this study are available in existing publications or upon request from the corresponding authors.

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

## Acknowledgements
We would like to acknowledge the memory of our colleague Dr. Barbara Eichhorn. Her archaeobotanical research in West and Central Africa continues to inspire. Our deepest thanks go to Manfred Eggert who was director of previous fieldwork conducted at BLD, IMB, and LON, and to the Deutsche Forschungsgemeinschaft for funding these projects. H.-P.W. and H.E. also acknowledge generous funding by Deutsche Forschungsgemeinschaft. M.B., N.B., and P.R. would like to thank the Max Planck Society for support and funding for this research. B.E. would like would like to express her deepest gratitude to Katharina Neumann for the constant support of her work, and Jennifer Markwirth for laboratory assistance. H.-P.W. wishes to express his gratitude to the population, the village council, and the chiefs of BLD (Tshuapa Province, DRC) for generously hosting archaeological field teams in 1983 and 2016, and for providing invaluable services. Mercader's team expresses his enormous gratitude to the Ituri Project, which operated in the Northeast region of the DRC since 1980 through the early 1990s. We are indebted to our Efe and Lese neighbours from Ngodingodi-Malembi, who hosted us so graciously and made this work possible.

## Author contributions
M.B. and P.R. designed the study. H.-P.W., B.E., J.M., and V.L. provided materials and provenancing information. M.B., J.Z., S.M., B.F., and P.R. performed isotope analysis of human and faunal remains. A.S. performed isotopic analysis of charred food remains. J.M., M.S., J.I., and S.C. facilitated and conducted dental calculus analysis. M.B., P.R., and H.-P.W. wrote the manuscript with input from all authors.

## Funding

## Competing interests
The authors declare no competing interests.
