## [Peer Review File · Communications Biology]

Reviewers' comments:

Reviewer #1 (Remarks to the Author):

This work represents a nice multidisciplinary study on the beginnings of agriculture in Central Africa. It also rectifies previous assumptions of modern crop species representing also key species in past societies.

Based on human, faunal and charred food remains from 4 archaeological sites, their results lead the authors to conclude a high diversity in subsistence practices, and a much lower C4 crop contribution to human diet than previously thought, during the Central African Iron Age.

For a topic such as the study of early agriculture multiple lines of evidence are required to move beyond common sense assumptions that are heavily based on indirect information sources such as genetic and linguistic models.

In contrast to the stable isotope data the microbotanical results are not really conclusive, but indicate a potential that needs to be further developed by future studies.

There should be some sentences on contamination issues, and how they can be excluded.

Reviewer #2 (Remarks to the Author):

This is an important study that provides much needed empirical information concerning the likely diets of Early Farming and Later Farming Communities within the Congo basin, and especially on the relative proportion of C4 to C3 foods in these diets.

I recommend publication with some minor corrections, especially to how the research results are presented and the discussion section.

The study provides the results of a suite of overlapping scientific analyses that tend to support one another, but also raise some further questions. The methods used are entirely appropriate and the analyses have followed recognised international standards and protocols. The supplementary data are clear and add depth and texture to the study.

The findings are novel and certainly advance understanding of the spread of food production within the Congo basin which, like so many other parts of sub-Saharan Africa, has been hindered by an over-reliance on material proxies (especially pottery types) for inferring evidence for food production rather than direct archaeobotanical evidence and bioarchaeological indices - as discussed here.

I fully concur also with the authors' statement (page 12, lines 278-9) that there is a need to shift away from 'broad linguistic and genetic models for 'the Bantu expansion'', and their results certainly provide support for this position.

BUT a) isn't this rather old hat? Several researchers have been making this very same point for close on two decades. I'd like to see you rephrase these kinds of statement to indicate that your empirical results underline the importance of the more theoretical points other scholars have been making for some time. And here this must also entail a questioning of the continuing relevance of the terms 'Bantu expansion' and 'Iron Age', and even efforts to tie evidence for changes in material culture and diet to linguistic changes - much better, in my view, to adopt the less loaded terms Early and Later Farming Communities, given the generic problem that linguists, archaeologists and geneticists are studying rather different (albeit potentially related) phenomena when they use their data-sets.

b) Given the relatively small size of your sample, I'd encourage more cautious language - as an example on page 12, lines 272-6 you make the suggestion that millet may have been used as a special purpose food, especially in feasts/ceremonies, rather than an everyday staple. This is

indeed possible, but I don't consider you have enough evidence to say (as you do) on page 12, line 275, that millet was 'actually primarily used in feasting' - change to possibly or potentially.

c) Perhaps my harshest comment - reading your discussion section I felt there was a bit of a strawman/person element to how you are presenting your data. A case in point is the opening sentence to this section (page 11, lines 252-3). You cite three sources as evidence that 'it has previously been argued that the initial Iron Age of Central Africa represented a sweeping arrival of Bantu speaking communities with cereal crops such as pearl millet' - which prompted me to ask myself, has the first author actually read any of these papers?

In the first place, neither Hiernaux nor Vansina mention millet (or even cereal crops) in their paper, and Russell et al mention millet only once. More importantly, none of these authors argued for sweeping transformations or seemingly large scale demic migration. Vansina's 'wave model' was an explicit attempt to move away from precisely such thinking, and he offers a sophisticated model of language change that goes well beyond demic migration as being the sole or even primary driver of change. Hiernaux's paper is quite old and superseded by more recent work, but even his study sought to 'complicate' earlier models. Russell et al seek to provide a model of change in material culture based on available radiocarbon dates - they are cautious about linking this to either linguistic changes in diet or the spread of farming, although they do note that ceramics have been used as a proxy indicator for all three. My point is that you don't need to position your research findings in such an antagonist way - especially when those you purport to critique appear to hold very similar positions on these issues.

My final point is less about the content of your paper or how you present your results, and more about your efforts to position your paper as having much wider relevance than to those with interests in the archaeology and linguistic and population histories of central Africa (which to my mind are all very worthy of study). Viz. you open and close with statements that seemingly position your paper and results as having a lot to contribute to addressing contemporary global challenges arising from current escalating climate change. I agree that the results might have relevance, and it is important that you make this point, but this can and should be made in a concluding paragraph - so move the opening lines down to here. Then write a different paper that goes into much greater depth about the applied value of this kind of research, since you do not cover that aspect at all in the paper.

Reviewer 3:

Review: Isotopic and Microbotanical Insights into Iron Age agricultural reliance in the Central African rainforest

The paper is generally interesting and important as it covers Central African Iron Age which is generally under-represented in the African Archaeology literature. However, the paper leaves some loose ties that make it difficult to convince the reader of the conclusions made.

1. There is no clear indication of the ceramic cultures or any other cultural affinities of the sites where samples come from. It is important to provide a detailed description of the (ceramic) cultural affinity of the sites studied and compare to those to other archaeological cultures in Central Africa.

2. In the main text, there is no clear description of the actual number of human individuals as well as animal samples included in the study. Moreover, it is not clear if the human remains come from relatively similar or different radio carbon dates/periods and therefore the discussion tends to be without a solid foundation since the information needed to anchor it is not there. Even the dated for the food remains at one of the sites is difficult to comprehend given that the authors have not given the relative context or dated of the food remains versus the human remains from the same site.

3. Of the animal samples used, how many are C4 and how many are C3 dependents? How many are grazers, how many are browsers and how many are mixed feeders? There is no demonstration of the C4 versus C3 vegetation composition of the region under study. And if possible can the authors indicate what the paleo-environment at the time of early occupation of the region. A general paleo-environmental context will still be fine.

4. The paper needs an overall re-arranging of sections. For example, the results section needs to go after materials and methods. The methods used for analysis are well presented but unfortunately the materials (number of humans per site, number of animals per site and their species) needs to be articulated in a clear and simple section.

5. In the discussion, what is the impact of the distances between the sites on the overall patterns of stable isotope values from humans?

In summary I think this paper needs to be published because it has very important foundational information on the archaeology of central Africa, which is presently missing in our knowledge of African archaeology. With revisions and polishing it will be a worthy paper for students and researchers in the continent. I do appreciate and understand that there may be a lot of missing in that part of the world, but it is always important that whatever little work that has been done before be acknowledged and linked to the current study. It is for this reason that I think the authors need to dig for any publications on central African archaeology, some of which may not be in English, unfortunately. Perhaps some links can be made with east Africa since the area under study is not far from the Great Lakes region where a lot Bantu migration archaeology has been studied.

Point-by-Point Response to Reviewers

We would like to thank the Reviewers for their highly constructive comments in relation to our manuscript. We are glad that they all recognised the importance and quality of our data and conclusions. We thank them also for their detailed suggestions to improve our manuscript. We have responded to these on a point-by-point basis below as well as in a word document with tracked changes and believe that they have greatly improved our article.

Reviewer #1 (Remarks to the Author):

This work represents a nice multidisciplinary study on the beginnings of agriculture in Central Africa. It also rectifies previous assumptions of modern crop species representing also key species in past societies. Based on human, faunal and charred food remains from 4 archaeological sites, their results lead the authors to conclude a high diversity in subsistence practices, and a much lower C4 crop contribution to human diet than previously thought, during the Central African Iron Age. For a topic such as the study of early agriculture multiple lines of evidence are required to move beyond common sense assumptions that are heavily based on indirect information sources such as genetic and linguistic models. In contrast to the stable isotope data, the microbotanical results are not really conclusive, but indicate a potential that needs to be further developed by future studies.

We thank the Reviewer for raising these points. We are glad that they recognise the benefits of our multidisciplinary approach and agree that our isotope data is conclusive, especially in providing a context-specific approach for overturning some previous assumptions and stereotypes based on indirect information sources. With regards to their concerns in relation to the microbotanical data we agree that development in future studies will represent a highly productive avenue for research in the region

There should be some sentences on contamination issues, and how they can be excluded.

For the isotope results, we have now added lines 117-120:

“Bone collagen results were assessed using established indicators of preservation, including a C/N ratio between 2.9-3.6, %C of ca.15-48%, and %N of ca. 5-17%⁴⁴⁻⁴⁷. Two results generated for human burials from BLD were excluded from final analysis as they produced a %N <5% and a %C <15% (Supplementary Table 3).”

In Supplementary Table 3 (unamended) samples that were excluded from final results due to having %N <5% and %C <15% are shown by an asterisk so that even our negative data is shown. We have now made this clear in the main text.

For the tooth enamel we have now also added lines 170-172:

“Tooth enamel is widely regarded as the archaeological material of choice in the tropics. Stable carbon and oxygen isotopes of tooth enamel have been shown to robustly preserve ecological variation, even in tropical regions, from the Miocene to the Late Pleistocene^{34,50,51}.”

For microbotanical remains, we have now added a sentence (220-222) about contamination concerns:

“The calcium phosphate matrix was decontaminated prior to decalcification, as per published protocol⁵⁴ in which calculus is immersed in sodium hydroxide of 2% w/v solution for 24h.”

Lines 222-224 (unamended) also give additional details about why the starch is believed to be ancient:

“As expected for ancient starch granules, the discovered calculus starch displays signs of damage to their semicrystalline matrix, having partially or totally lost their native birefringence. Other signs of diagenesis include fissuring, pitting, granulation, and implosion of the hilum.”

We also added the following (lines 446-450) to the Methods:

“We present microbotanical materials released from the calcified matrix after thorough decontamination protocols and decalcification in a cleanroom laboratory (as per⁵⁴), as well as microbotanicals still trapped in the calculus matrix but visible enough to have their two dimensional morphology identified. We followed the morphometric classification criteria for the identification of ancient starch from sub-Saharan plants in⁵⁵.”

Reviewer #2 (Remarks to the Author):

This is an important study that provides much needed empirical information concerning the likely diets of Early Farming and Later Farming Communities within the Congo basin, and especially on the relative proportion of C4 to C3 foods in these diets. I recommend publication with some minor corrections, especially to how the research results are presented and the discussion section. The study provides the results of a suite of overlapping scientific analyses that tend to support one another, but also raise some further questions. The methods used are entirely appropriate and the analyses have followed recognised international standards and protocols. The supplementary data are clear and add depth and texture to the study. The findings are novel and certainly advance understanding of the spread of food production within the Congo basin which, like so many other parts of sub-Saharan Africa, has been hindered by an over-reliance on material proxies (especially pottery types) for inferring evidence for food production rather than direct archaeobotanical evidence and bioarchaeological indices - as discussed here.

We thank the Reviewer for their positive comments in relation to the novelty importance of our manuscript and its contribution of much needed empirical information in relation to the diets of farming communities within the Congo Basin. We are glad that they approve of our methodological approach and find out supplementary data clear.

I fully concur also with the authors' statement (page 12, lines 278-9) that there is a need to shift away from 'broad linguistic and genetic models for 'the Bantu expansion', and their results certainly provide support for this position.

We are also glad that the Reviewer agrees with our conclusion in relation to the need for direct dietary data in relation to the 'Bantu expansion' and we hope that more studies will continue in this vein.

BUT a) isn't this rather old hat? Several researchers have been making this very same point for close on two decades. I'd like to see you rephrase these kinds of statement to indicate that your empirical results underline the importance of the more theoretical points other scholars have been making for some time.

We thank the reviewer for their detailed feedback. We have expanded the introduction (lines 40-68) to acknowledge previous efforts to deconstruct sweeping models for the 'Bantu Expansion' as follows:

*"For the last half a century, if not longer, the processes for the dispersal of Bantu-speaking communities from Western Central Africa have been a major focus of African archaeological, linguistic, and genetic research¹⁻⁴. **While there has been an increasing departure from notions of a single sweeping "Bantu Expansion", the degree to which the movement of people, languages, and the emergence of farming are linked across Africa continues to be forcefully debated⁵⁻⁷. Central Africa is at a key location for developing existing models for the spread of farming⁸ yet investigations of the emergence of food production, particularly in the rainforest, have been limited⁹. Assumptions that tropical rainforests represent significant barriers to agriculturalists¹⁰ have been used to rationalise a relatively late arrival of farming in the region, c. 2,500 years ago, during a period of climate- or human-induced deforestation¹¹⁻¹³ (Supplementary Text 1). However, unlike other parts of Africa¹⁴⁻¹⁶, there have been few studies directly testing changes in human dietary reliance on agricultural crops, relative to local freshwater, bushmeat, and tropical forest plant resources¹⁷, from the first arrival of domesticates in the region through to the present day.***

*Linguistic, material culture, and radiocarbon analyses have now shown that human arrival throughout the Congo Basin was a complex and time-transgressive occurrence, potentially with the interaction of different populations occurring occurring^{10,18-25}. Furthermore, ideas relating to the inability of farming populations to occupy the tropical rainforests of Central Africa have come under renewed scrutiny²⁶. Experimental research has demonstrated that pearl millet (*Pennisetum glaucum*) can be grown in forested portions of the Inner Congo Basin²⁷. This suggests that discoveries of pearl millet (c. 2330-330 BP) at Iron Age sites across Central Africa, regions presently covered in tropical rainforest^{18,28}, need not represent a time of mass*

'rainforest crisis'^{29,30}. Not only that, but Iron Age expansions into the various tributaries of the Congo River continued well after the supposed peak in rainforest decline 2,500 years ago, suggesting more complex, ongoing processes of agricultural adaptation and settlement. Together, these developments make it essential to build more integrated, multidisciplinary, and context-specific insights into changes in diet and land-use in different parts of Central Africa through time, as different agricultural populations negotiated their tropical surroundings."

And here this must also entail a questioning of the continuing relevance of the terms 'Bantu expansion' and 'Iron Age', and even efforts to tie evidence for changes in material culture and diet to linguistic changes - much better, in my view, to adopt the less loaded terms Early and Later Farming Communities, given the generic problem that linguists, archaeologists and geneticists are studying rather different (albeit potentially related) phenomena when they use their data-sets.

We thank the Reviewer for this comment in relation to terminology. We have added the following (lines 42-45) to the Introduction:

"While there has been an increasing departure from notions of a single sweeping "Bantu Expansion", the degree to which the movement of people, languages, and the emergence of farming are linked across Africa continues to be forcefully debated⁵⁻⁷."

Nevertheless, we retain the use of the terms 'Bantu expansion' and 'Iron Age' in order to make our manuscript more widely accessible to geneticists and linguists seeking to make more nuanced approaches to human history in the region. We hope this satisfies the Reviewer but are happy to change further should the Editor consider it necessary.

b) Given the relatively small size of your sample, I'd encourage more cautious language - as an example on page 12, lines 272-6 you make the suggestion that millet may have been used as a special purpose food, especially in feasts/ceremonies, rather than an everyday staple. This is indeed possible, but I don't consider you have enough evidence to say (as you do) on page 12, line 275, that millet was 'actually primarily used in feasting' - change to possibly or potentially.

Thank you for pointing this out. We have tried to make our language more cautious at the relevant points in the manuscript. In relation to this particular sentence (lines 297-299) we have changed as follows to:

*"This potentially implies that millet was used in a different context in the Iron Age of this region and, instead of being a staple, was **possibly** used in feasting, brewing, or prestige contexts"*

c) Perhaps my harshest comment - reading your discussion section I felt there was a bit of a strawman/person element to how you are presenting your data. A case in point is the opening sentence to this section (page 11, lines 252-3). You cite three sources as evidence that 'it has previously been argued that the initial Iron Age of Central Africa represented a

sweeping arrival of Bantu speaking communities with cereal crops such as pearl millet' - which prompted me to ask myself, has the first author actually read any of these papers?

In the first place, neither Hiernaux nor Vansina mention millet (or even cereal crops) in their paper, and Russell et al mention millet only once. More importantly, none of these authors argued for sweeping transformations or seemingly large scale demic migration. Vansina's 'wave model' was an explicit attempt to move away from precisely such thinking, and he offers a sophisticated model of language change that goes well beyond demic migration as being the sole or even primary driver of change. Hiernaux's paper is quite old and superseded by more recent work, but even his study sought to 'complicate' earlier models. Russell et al seek to provide a model of change in material culture based on available radiocarbon dates - they are cautious about linking this to either linguistic changes in diet or the spread of farming, although they do note that ceramics have been used as a proxy indicator for all three. My point is that you don't need to position your research findings in such an antagonist way - especially when those you purport to critique appear to hold very similar positions on these issues.

We thank the reviewer for their suggestions and comments and take their points. We have reframed our Introduction (see previous answer to point a) as follows:

“For the last half a century, if not longer, the processes for the dispersal of Bantu-speaking communities from Western Central Africa have been a major focus of African archaeological, linguistic, and genetic research¹⁻⁴. While there has been an increasing departure from notions of a single sweeping “Bantu Expansion”, the degree to which the movement of people, languages, and the emergence of farming are linked across Africa continues to be forcefully debated⁵⁻⁷. Central Africa is at a key location for developing existing models for the spread of farming⁸ yet investigations of the emergence of food production, particularly in the rainforest, have been limited⁹. Assumptions that tropical rainforests represent significant barriers to agriculturalists¹⁰ have been used to rationalise a relatively late arrival of farming in the region, c. 2,500 years ago, during a period of climate- or human-induced deforestation¹¹⁻¹³ (Supplementary Text 1). However, unlike other parts of Africa¹⁴⁻¹⁶, there have been few studies directly testing changes in human dietary reliance on agricultural crops, relative to local freshwater, bushmeat, and tropical forest plant resources¹⁷, from the first arrival of domesticates in the region through to the present day.

Linguistic, material culture, and radiocarbon analyses have now shown that human arrival throughout the Congo Basin was a complex and time-transgressive occurrence, potentially with the interaction of different populations occurring occurring^{10,18-25}. Furthermore, ideas relating to the inability of farming populations to occupy the tropical rainforests of Central Africa have come under renewed scrutiny²⁶. Experimental research has demonstrated that pearl millet can be grown in forested portions of the Inner Congo Basin²⁷. This suggests that discoveries of pearl millet (*Pennisetum glaucum*) (c. 2330–330 BP) at Iron Age sites across Central Africa, regions presently covered in tropical rainforest^{18,28}, need not represent a time of mass ‘Rainforest Crisis’^{29,30}. Not only that, but Iron Age expansions into the various tributaries of the Congo River continued well after the supposed peak in rainforest decline 2,500 years ago,

suggesting more complex, ongoing processes of agricultural adaptation and settlement. Together, these developments make it essential to build more integrated, multidisciplinary, and context-specific insights into changes in diet and land-use in different parts of Central Africa through time, as different agricultural populations negotiated their tropical surroundings.”

We have also changed the opening of our discussion (lines 271-276):

*“The discovery of pearl millet (*Pennisetum glaucum*) in Iron Age pits in Southern Cameroon¹⁸ was one of the most significant findings of the past two decades in Central African archaeology. The discovery sparked debate over the environmental context for early agriculture^{26,56}, **contributing to an increasingly complex narrative for the settlement of the Central Africa rainforest^{10,21,57,58}**. Yet direct assessments about the degree to which these early farming communities, particularly in the DRC, relied on C₄ sources are limited¹⁷.”*

We hope that the Reviewer now feels that this does just treatment to the works suggested by the Reviewer and also provides a more nuanced introduction to what can be controversial subject matter.

My final point is less about the content of your paper or how you present your results, and more about your efforts to position your paper as having much wider relevance than to those with interests in the archaeology and linguistic and population histories of central Africa (which to my mind are all very worthy of study). Viz. you open and close with statements that seemingly position your paper and results as having a lot to contribute to addressing contemporary global challenges arising from current escalating climate change. I agree that the results might have relevance, and it is important that you make this point, but this can and should be made in a concluding paragraph - so move the opening lines down to here. Then write a different paper that goes into much greater depth about the applied value of this kind of research, since you do not cover that aspect at all in the paper.

We thank the Reviewer for this point. For the Introduction, as can be seen from our responses to two of the Review’s points above we have now completely and removed statements about contemporary global challenges of food security in the region from the Introduction. As an alternative, we have focused on the archaeological context and debates surrounding the emergence of food production in the region.

Instead, we have now focused our discussions of global relevance into the final concluding paragraph (lines 339-345) as follows:

*“In this way we can also begin to properly understand the wider significance of ongoing agricultural adaptations in Central Africa. The discovery of pearl millet (*Pennisetum glaucum*) in Iron Age pits in Southern Cameroon¹⁸ was one of the most significant findings of the past two decades in Central African archaeology. The discovery sparked debate over the environmental context for early agriculture^{26,56}, **contributing to an increasingly complex narrative for the settlement of the Central Africa rainforest^{10,21,57,58}**. Yet direct assessments about the degree to which these early farming communities, particularly in the DRC, relied on C₄ resources are limited¹⁷. Despite frequent NGO or government calls to focus on productive cereal monoculture*

to meet growing populations in West and Central Africa⁷⁶, it is clear that mixed use of C₄ plants, wild resources, rainforest economic plants, and freshwater resources have characterised subsistence practices in the western Democratic Republic of the Congo for over 2,000 years. Indeed, the continuation of heterogeneous food production strategies could prove crucial in ensuring long-term food security in tropical Central Africa, as well as the survival of Congolese environments crucial to the global carbon cycle⁷⁷ pan-African precipitation⁷⁸, and global biodiversity⁷⁹.”

We hope that they are satisfied by our changes.

Reviewer #3 (Remarks to the Author):

The paper is generally interesting and important as it covers Central African Iron Age which is generally under-represented in the African Archaeology literature. However, the paper leaves some loose ties that make it difficult to convince the reader of the conclusions made.

We are glad that the Reviewer found our paper of interest and agree that it makes an important contribution to the Central African Iron Age and its position within discussions of African Archaeology more widely. In relation to their concerns, we hope to have addressed them on a point-by-point basis below.

1. There is no clear indication of the ceramic cultures or any other cultural affinities of the sites where samples come from. It is important to provide a detailed description of the (ceramic) cultural affinity of the sites studied and compare to those to other archaeological cultures in Central Africa.

We thank the Reviewer for this comment. We have now added further details in relation to this point in the Introduction as follows (lines 75-87):

“IMB is the type-site for the earliest pottery tradition of the central equatorial rainforest and the individual analysed, indirectly dated to around 2,050 BP (Supplementary Text 2), would have been a member of a group representing already established agriculture in the region after its initial settlement by sedentary immigrant populations a few centuries earlier. In contrast, individuals from LON and BLD represent subsistence practices during the Late Iron Age when populations were spreading further across the Congo Basin. Finally, isotopic results from the individual from MTNW, previously identified as a likely hunter-gatherer³¹ offers new evidence about the intricacies of subsistence, cultural and genetic identities further to the eastern edge of the Basin (Supplementary Text 2). Collectively, the samples analysed span the period following the first arrival of food producers in this region (~2,050 BP) through to relatively recent occupation (~130 BP) (Supplementary Text 2, Supplementary Tables 1-2, Supplementary Fig. 1, 3–7).”

Further details about the chronological and cultural context of the sites are in Supplementary Text 2 as follows:

Imbonga:

*“Imbonga (IMB) is a waterside village located on the Momboyo River. **It is the type-site for the earliest pottery tradition of the central equatorial rainforest.** Several excavations were carried out here in the 1980s by Manfred Eggert^{30–32}, and a number of radiocarbon dates were published (Supplementary Table 1). **For the present study, enamel was sampled from a human second molar discovered in sediment contained in a ceramic vessel at IMB 81/11 (Supplementary Table 1).** It is a second molar of an individual between 9–12 years of age. No other skeletal remains were found in this context. **The vessel, a richly decorated flat-based bowl attributable to the Early Iron Age Inganda style, had been found isolated but in spatial proximity to a number of Early Iron Age pottery deposits. Although possibly not in situ at discovery, the vessel was found upside down, i.e. in a position typical of Early Iron Age ritual pottery deposits known from the Inner Congo Basin. Two holes intentionally knocked into opposite sides of the vessel wall likewise suggest ceremonial burial. IMB 81/11 has not been directly dated. However, Inganda pottery is known to have been produced in the 2nd and 1st centuries cal. BC.**”*

Assuming contextual integrity of the find, the tooth sample may therefore be regarded as by far the oldest evidence included in this study representing one of the earliest periods of the regional Early Iron Age, 200 years more recent, at the most, than regional Iron Age beginnings associated with Imbonga ceramics. Imbonga pottery has been found at some 60 sites across the western parts of the Inner Congo Basin. It is not associated with any stone artefacts³² but with a sedentary way of life, an advanced iron metallurgy, and a food production system.”

Matangai Turu Northwest:

*“Contextually, the human remains associate with LSA lithics⁴¹, **Late Iron Age ceramics stylistically connected to those from the Western branch of the East African Rift System²**, and one isolated iron bar, without evidence of smelting or forging⁴³.”*

We have added the following lines to Supplementary Text 2 about Bolondo:

“None of the Bolondo burials were associated with ceramic finds interpretable as grave goods. However, all of them were found in stratigraphic contexts representing intermediate and recent layers of the site stratigraphy. These are characterised by settlement refuse including large quantities of pottery fragments belonging to the following style groups of the regional Later Iron Age Tshuapa Tradition (in stratigraphic sequence from earliest to most recent): Bolondo, Bokone, Bolombi, and Ilemba-Bokonda³².”

2. In the main text, there is no clear description of the actual number of human individuals as well as animal samples included in the study.

We have now added an explicit comment in relation to the number of individuals to the introduction (lines 71-74):

“Isotopic results were obtained for human burials from the sites of Imbonga (IMB) (n = 1), Longa (LON) (n = 1), Bolondo (BLD) (n = 18), and Matangai Turu Northwest (MTNW) (n = 1). In addition, bone collagen (n = 10) and enamel (n = 6) was analysed for a range of fauna from Bolondo to create an isotopic baseline.”

Moreover, it is not clear if the human remains come from relatively similar or different radiocarbon dates/periods and therefore the discussion tends to be without a solid foundation since the information needed to anchor it is not there. Even the date for the food remains at one of the sites is difficult to comprehend given that the authors have not given the relative context or date of the food remains versus the human remains from the same site.

We have now made this clear in the Introduction (lines 75-87):

“IMB is the type-site for the earliest pottery tradition of the central equatorial rainforest and the individual analysed, indirectly dated to around 2,050 BP (Supplementary Text 2), would have been a member of a group representing already established agriculture in the region after its initial settlement by sedentary immigrant populations a few centuries earlier. In contrast, individuals from LON and BLD represent subsistence practices during the Late Iron Age when populations were spreading further across the Congo Basin. Finally, isotopic results from the individual from MTNW, previously identified as a likely hunter-gatherer³¹ offers new evidence about the intricacies of subsistence, cultural and genetic identities further to the eastern edge of the Basin (Supplementary Text 2). Collectively, the samples analysed span the period following the first arrival of food producers in this region (~2,050 BP) through to relatively recent occupation (~130 BP) (Supplementary Text 2, Supplementary Tables 1-2, Supplementary Fig. 1, 3-7).”

Further information about the sites can be found in Methods and Supplementary Text 2 (unamended).

3. Of the animal samples used, how many are C4 and how many are C3 dependents? How many are grazers, how many are browsers and how many are mixed feeders?

We have now added totals to lines 121-126 of the results as follows:

“The fauna from BLD fall broadly into four groups: wild browsers (antelope and duiker, n = 2), domesticated browsers (goats, n = 2), mammalian carnivores (n = 2), and aquatic species (fish and crocodile, n = 4). $\delta^{13}\text{C}$ values from BLD mammals (n = 6) are consistent with a largely C₃-based diet with measurements ranging from -23.7‰ to -18.2‰, although the goats, the dog and the fox-sized carnivore could potentially have some C₄ component to the diet (Fig. 2) (Supplementary Table 3).”

There is no demonstration of the C4 versus C3 vegetation composition of the region under study. And if possible can the authors indicate what the paleo-environment at the time of early occupation of the region. A general paleo-environmental context will still be fine.

We have added the following lines (88-105) to the Introduction as follows to provide palaeoenvironmental context:

*“ $\delta^{13}\text{C}$ analysis of human tissues has long been demonstrated to provide insights into reliance on plants with different photosynthetic pathways (namely C_4 versus C_3) and their animal consumers (Supplementary Text 3)³²⁻³⁴. Significantly, in Central Africa, wild, as well as potentially domesticated (e.g. yams), forest plants are C_3 , while incoming cereal crops (e.g. pearl millet, sorghum and, for later periods, maize) are C_4 . **$\delta^{13}\text{C}$ measurements of wild plants and animals from the rainforests of the Democratic Republic of Congo show that these forests are largely composed of C_3 vegetation³⁵. Moreover, they show a recognisable ‘canopy effect’ on this C_3 vegetation that results in lower $\delta^{13}\text{C}$ among plants, and their animal consumers, living under dense canopies compared to those living in more open areas³⁵, something that has been well-documented in many other tropical regions (e.g.^{34,36}).**”*

*The sites of IMB, LON and BLD are located on tributaries of the Congo River in the western DRC (Figure 1, Supplementary Text 2) **an area presently covered in dense C_3 -dominated evergreen and semi-deciduous forest³⁷. Stable carbon measurements of faunal tooth enamel from BLD, which largely reflect the proportions of C_3/C_4 plants consumed, reflect local palaeoecology as well as providing baseline values for human diet. The final site, MTNW, is situated in the closed-canopy forest of the Ituri Region of the Northeast Congo Basin with palaeoenvironmental proxies suggesting a predominance of tropical forest tree taxa during the time of occupation^{38,39}.**”*

A broader summary of the palaeoenvironmental context for Central Africa during the Holocene is also presented in Supplementary Text 1 (unamended):

“There have been, however, significant counter-arguments against a strong human impact on the forests during this phase, in particular, a discrepancy between timing of clear settlements and large-scale vegetation changes^{11,12}.

It has been proposed that the opening up of a ‘Savannah Corridor’ c. 2,500 cal. BP facilitated the movement of Bantu speaking farmers, from their hypothesised homeland at the present-day Nigeria-Cameroon border area, eastwards and southwards across the rest of the African continent^{3,10,13}. Since the opening up of the landscape has commonly been considered to have supported the introduction and cultivation of new crops, the equatorial forest of Central Africa has been viewed as a barrier for agricultural expansion.”

However, here we have also added further clarification about C_3 vs C_4 plants in Supplementary Text 1:

“The paucity of domestic C_4 cereal finds at African rainforest sites makes it challenging to explore this archaeologically however, and the importance of forest C_3 crops, including yams and fruit

trees such as *Canarium*, and of oil palm (*Elaeis guineensis*) exploitation, should not be overlooked^{1,22-24}. **Charred endocarps of *Canarium schweinfurthii* recovered from rockshelters in the Ituri rainforest attest to consumption of wild fruits throughout the Holocene²⁴. Oil palm endocarp remains in particular, are abundant at rainforest sites²⁵, and likely provided a major source of fat.** Tubers such as wild and domesticated yams are still difficult to trace in archaeobotanical records though ethnographic evidence shows the importance and efficiency of wild yams exploitation for human nutrition in the Central African rainforest (e.g.^{26,27}), including species which also occur in the Inner Congo Basin. The potential role of domesticated yams during the Iron Age so far remains unresolved. Furthermore, archaeobotanical remains only show detailed snapshots of specific plants, in contrast, the isotope analysis of human and faunal tissue can provide an assessment of overall dietary reliance."

We have also added further details to Supplementary Text 3 as follows:

" $\delta^{13}\text{C}$ and $\delta^{15}\text{N}$ analysis of bulk bone collagen is one of the most commonly applied approaches, and enables exploration of the degree to which humans were reliant on plants following the two main photosynthetic pathways (C_3 and C_4), and their animal consumers, as well as their position within the foodchain. **The archaeological sites of this study currently represent tropical forest environments largely dominated by C_3 forest species, as well as oil palm (*Elaeis guineensis*). Phytolith evidence for MTNW has confirmed a similar environmental context (closed canopy forest) was present at the time of occupation⁴⁰.** These C_3 plants that dominate tropical forest environments, as well as domesticated oil palm and yams, have a $\delta^{13}\text{C}$ between -35‰ and -19‰ making them distinguishable from C_4 plants, which include wild grasses, millets and sorghum, which have a $\delta^{13}\text{C}$ between -13‰ to -8‰⁴⁷⁻⁵⁰. C_3 plants growing under a closed canopy, as well as their consumers, have even lower $\delta^{13}\text{C}$ due to low light and recycled the CO_2 the well-documented so-called 'canopy effect'^{51,52}. The distinct $\delta^{13}\text{C}$ of these groups of plants is passed into the tissues of their consumers with a known fractionation effect⁵⁴."

This information all comes together to show the C_4 versus C_3 vegetation distributions. Namely, the tropical forests are dominated by C_3 plants, as elsewhere in the tropics, meanwhile in grassland patches, and as grasslands open up during drier periods, C_4 grasses and sedges can appear, as documented in palaeoenvironmental studies in the region (Garcin et al. 2018).

4. The paper needs an overall re-arranging of sections. For example, the results section needs to go after materials and methods.

In accordance with formatting guidelines for *Communications Biology*, the Methods are presented after the results. This is standard practice for this journal so we have left it where it is.

The methods used for analysis are well presented but unfortunately the materials (number of humans per site, number of animals per site and their species) needs to be articulated in a clear and simple section.

As above, we have added this information into the Introduction. We have also added these details at the beginning of the Methods section (lines 354-366) as follows:

“Four Iron Age sites were selected for study from across the DRC (Fig. 1, Supplementary Text 2): Imbonga (IMB), Longa (LON), Bolondo (BLD) and Matangai Turu Northwest (MTNW). Bulk bone collagen $\delta^{13}\text{C}$ and $\delta^{15}\text{N}$ measurements were obtained for human burials from LON ($n = 1$), BLD ($n = 11$), and MTNW ($n = 1$). Totals reflect results taken forward for interpretation. For BLD, two results were excluded from final analysis due to poor preservation (for full details see: Results, Supplementary Table 3). Bone collagen was also analysed from a range of faunal remains from BLD ($n = 10$) to establish a dietary baseline, including domestic browsers (goats), wild browsers (duiker) and fish. To further explore dietary intake, human tooth enamel was sampled from IMB ($n = 1$), LON ($n = 1$), BLD ($n = 11$) and MTNW ($n = 1$). Due to differential preservation, it was only possible to generate both a bulk collagen and tooth enamel results for four human burials from BLD and the single individual from LON. Additionally, animal tooth enamel from BLD ($n = 6$) was analysed for $\delta^{13}\text{C}$ and $\delta^{18}\text{O}$ to aid the interpretation of human values.”

5. In the discussion, what is the impact of the distances between the sites on the overall patterns of stable isotope values from humans?

The three sites of LON, IMB and BLD are relatively close to one another, at air-line distances of c. 120, 170 and 270 km, respectively. They are so similar in terms of present ecological setting, hydrology, and human food resources that one would not anticipate significantly different isoscapes. We recognise that MTNW is in a very different location from the other sites (mean distance c. 1050 km as the crow flies) and have tried to treat this individually separately both in our results and discussion. We believe we have now made the difference in social and environmental context very explicit in the main text and SI.

For example, we have attempted to acknowledge how the location of MTNW could influence the dietary signal we observed. Lines 314-325 (unamended):

“The location of Matangai Turu in proximity to migrating farming populations in eastern Africa⁶², observed genetic affinity of the sampled individual to these groups as well as hunter-gatherer populations⁶³, and the identification of Poaceae starch (likely sorghum) in the dental calculus of this individual indicate that this C_4 signal represents the use of sorghum possibly acquired through interactions with agricultural groups in eastern Africa. Nevertheless, the human remains date to 813 +/- 35 BP, a time when the region is believed to have been covered by lowland Guineo-Congolain rainforest^{31,35,38,64}. Moreover, the presence of starches from Dioscoreaceae and Asphodelaceae in the dental calculus, as well as the association with wild rainforest fauna^{54,64} indicate ongoing contributions of forest resources to the lifeways of this Late Iron Age population. Evidently, the adoption of cereal crops into subsistence economies across tropical Africa was not uniform, displaying regional variation and incorporation into existing, dynamic lifeways.”

We acknowledge their different temporal contexts and have clarified this in the Introduction (lines 75-87):

“IMB is the type-site for the earliest pottery tradition of the central equatorial rainforest and the individual analysed, indirectly dated to around 2,050 BP (Supplementary Text 2), would have been a member of a group representing already established agriculture in the region after its initial settlement by sedentary immigrant populations a few centuries earlier. In contrast, individuals from LON and BLD represent subsistence practices during the Late Iron Age when populations were spreading further across the Congo Basin. Finally, isotopic results from the individual from MTNW, previously identified as a likely hunter-gatherer³¹ offers new evidence about the intricacies of subsistence, cultural and genetic identities further to the eastern edge of the Basin (Supplementary Text 2). Collectively, the samples analysed span the period following the first arrival of food producers in this region (~2,050 BP) through to relatively recent occupation (~130 BP) (Supplementary Text 2, Supplementary Tables 1-2, Supplementary Fig. 1, 3-7).”

We have also added more information to the Introduction about the importance of faunal data in interpreting the results (lines 100-102):

“Stable carbon measurements of faunal tooth enamel from BLD, which largely reflect the proportions of C₃/C₄ plants consumed, reflect local palaeoecology as well as providing baseline values for human diet.”

In summary I think this paper needs to be published because it has very important foundational information on the archaeology of central Africa, which is presently missing in our knowledge of African archaeology. With revisions and polishing it will be a worthy paper for students and researchers in the continent. I do appreciate and understand that there may be a lot of missing pieces in that part of the world, but it is always important that whatever little work that has been done before be acknowledged and linked to the current study. It is for this reason that I think the authors need to dig for any publications on central African archaeology, some of which may not be in English, unfortunately. Perhaps some links can be made with east Africa since the area under study is not far from the Great Lakes region where a lot Bantu migration archaeology has been studied.

We thank the Reviewer for their highly positive and encouraging remarks. We hope to have polished the paper to their satisfaction. In relation to the wider reading, following Reviewer 1's comments we have added further references to existing archaeology in the region and also refined our arguments following Reviewer 2's comments. We have also referenced more volumes and papers in German and French (e.g. Eggert 2012, Wotzka 1995, 2019, Oslisly et al. 2016). We thank the Reviewer again for their highly constructive review.

REVIEWERS' COMMENTS:

Reviewer #1 (Remarks to the Author):

I have read through the revised manuscript and I see that the authors have attended to all comments I had raised. The paper is now much easier to follow and they have removed some unnecessary details that were there in the first submission. The only editorial issue i picked was an incomplete sentence on line 483, "Samples were treated". They can fix it in just a second and that is all.

Reviewer #3 (Remarks to the Author):

This is an improved version of the paper. The authors have addressed all the concerns I raised in my original review to my satisfaction.

I noticed one typographic error - on line 2007 I suggest the authors insert the word 'the' before 'individual'

I am happy to recommend the paper for publication as it stands.

Point-by-Point Response to Reviewers

We would like to thank the Reviewers for their final comments in relation to our manuscript, please see our responses below.

Reviewer #1 (Remarks to the Author):

I have read through the revised manuscript and I see that the authors have attended to all comments I had raised. The paper is now much easier to follow and they have removed some unnecessary details that were there in the first submission. The only editorial issue I picked was an incomplete sentence on line 483, "Samples were treated" . They can fix it in just a second and that is all.

We thank the Reviewer for their feedback. The sentence now reads: "*Samples were pretreated using previously published methods*⁸⁶."

Reviewer #3 (Remarks to the Author):

This is an improved version of the paper. The authors have addressed all the concerns I raised in my original review to my satisfaction. I noticed one typographic error - on line 2007 I suggest the authors insert the word 'the' before 'individual'. I am happy to recommend the paper for publication as it stands.

We thank the Reviewer for their comments. We have inserted the word 'the' as suggested.